# “Smart Extraction Chain” with Green Solvents: Extraction of Bioactive Compounds from *Picea abies* Bark Waste for Pharmaceutical, Nutraceutical and Cosmetic Uses

**DOI:** 10.3390/molecules27196719

**Published:** 2022-10-09

**Authors:** Stefania Sut, Erica Maccari, Gokhan Zengin, Irene Ferrarese, Francesca Loschi, Marta Faggian, Bertoni Paolo, Nicola De Zordi, Stefano Dall’Acqua

**Affiliations:** 1Department of Pharmaceutical and Pharmacological Sciences, University of Padova, Via Marzolo 5, 35131 Padova, Italy; 2Department of Biology, Science Faculty, Selcuk University, 42130 Konya, Turkey; 3Unired Srl, Via Niccolò Tommaseo 69, 35131 Padova, Italy; 4Holz Pichler SpA, Ega—Stenk 2, Bolzano, 39050 Nova Ponente, Italy; 5Società Agricola Moldoi—S.A.M, SrL, Belluno, Loc. Maras Moldoi 151/a, 32037 Sospirolo, Italy

**Keywords:** *Picea abies* bark, green extraction, supercritical CO_2_, ultrasound assisted extraction, microwave assisted extraction, antioxidant, enzyme inhibition assays

## Abstract

Secondary metabolites from the sawmill waste *Picea abies* bark were extracted using an innovative two-step extraction that includes a first step with supercritical CO_2_ (SCO_2_) and a second step using green solvents, namely ethanol, water, and water ethanol mixture. Maceration (M), ultrasound assisted extraction (UAE) and microwave assisted extraction (MAE) techniques were applied in the second step. A total of nineteen extract were obtained and yield were compared. Bark extracts were characterized by LC-DAD-MS^n^ and classes of compounds were quantified as abietane derivatives, piceasides, flavonoids, and phenolics to compare different extractions. Obtained extracts were studied by in vitro assay to evaluate potential pharmaceutical, nutraceutical and cosmetic uses assessing the antioxidant activity as well as the inhibitory activity on target enzymes. Results show that the “smart extraction chain” is advantageous in term of yield of extraction and phytoconstituent concentration. SCO_2_ extract, presenting a unique composition with a large amount of abietane derivatives, exerted the best activity for amylase inhibition compared to the other extracts.

## 1. Introduction

Norway spruce (*Picea abies* (L.) Karst.) is a major softwood species in Europe extensively used in pulp mills for mechanical and kraft pulp production, and in sawmills to produce wood components. The industrial processing generates large quantities of biomass as residual materials that accumulate at mill site [1]. Timber industries nowadays use *P. abies* bark for fuel in combustion. In this work, a new extraction procedure is developed on *P. abies* bark sawmill waste, in compliance with the principle of the circular economy in which exhaustive recycling and re-use at every step of the production chain is sought [2]. 

*P. abies* bark is a complex plant matrix that contain different classes of secondary metabolites that can have useful application in cosmetic nutraceutical and pharmaceutical fields. Extraction can become an innovative step that allow the development of new products with unique characteristics. One of the main challenges to obtaining a high value extract from plant material is to apply innovative extraction techniques that avoid the utilization of organic solvent. ‘Green’ extraction techniques require less time, energy, and solvent, and are thereby in line with sustainable development strategies. The use of ‘green’ solvents led to products that are preferred by consumers and recognized as safe [3]. 

In this paper, *P. abies* bark material is extracted using different techniques in sequence following a scheme that we called “smart chain extraction”. The hypothesis at the basis of our work is to use the SCO_2_ extraction as a green approach to extract the resinous material from the bark. This first step should allow more easy penetration of solvent in the further step of extraction. At first, material is subjected to SCO_2_ extraction, in the second step, material is extracted using solvent-based extraction. For the second step, three different extraction techniques were compared, namely ultrasound assisted extraction (UAE), microwave assisted extraction (MAE), and maceration (M). Each of these extraction approaches have some unique characteristic [4].

SCO_2_ extraction can be a valuable technique for the extraction of lipophilic compounds, it avoids the use of solvents and allowed the extraction of lipophilic fractions in mild temperature conditions. Specifically, SCO_2_, due to its lower critical temperature (31 °C), produces extracts with fewer oxidation products [5]. 

SCO_2_ is lipophilic, useful for the extraction of non-polar compound. To obtain the extraction of polar secondary metabolites, we decided to perform a second step using green-solvents based on water and ethanol. Considering our previous work on *P. abies* bark, in this paper, extraction will be performed comparing water 100%, ethanol/water 50%, and ethanol 100%. 

M is the most used technique for the extraction from natural products, it is a very simple extraction method largely documented and with long tradition of use, with the disadvantage of long extraction time and low extraction efficiency, high solvent consumption. It could be used for the extraction of thermolabile components [4].

To improve extraction efficacy following sustainability approach, we compared maceration with UAE and MW. UAE is useful to obtain high valuable compounds from plant materials. Ultrasound in the solvent produces cavitation, accelerates the dissolution and diffusion of the solute, as well as the heat transfer, improving the extraction efficiency. The main benefits of UAE are related to energy consumption, namely the use of moderate temperatures, which is beneficial for heat-sensitive compounds. The highest extraction rate is usually achieved in the first few minutes, which is the most profitable period [6]. 

In MAE, microwaves generate heat by interacting with polar compounds such as water due to the ionic conduction and dipole rotation mechanisms. The transfers of heat and mass are in the same direction in MAE, generating a synergistic effect to accelerate extraction improving extraction yield. MAE application provides many advantages, such as increasing the extract yield, decreasing the thermal degradation [4]. 

In this paper *P. abies* bark sawmill is subjected to a “smart chain extraction” with a sustainable and green approach to obtain high value extract rich in secondary metabolites. The bark after the extraction in SCO_2_ and raw plant material (without the first extraction) are subjected to solvent extraction using M, UAE and MAE. Extracts are characterized by LC-DAD-MS^n^ and chemical composition in secondary metabolites is compared to establish if the SCO_2_ improve the further step of extraction. The activity of obtained extracts is investigated using in vitro antioxidant (radical scavenging, reduction ability and metal chelating) and enzyme inhibitory (cholinesterase, tyrosinase, amylase and glucosidase) assays, to valorize the possible product of the extraction of sawmill waste. 

## 2. Results

A schematic representation of the work is reported in Figure 1, the same batch of bark is used to compare the sequential extraction scheme of the “smart chain extraction” with the extraction obtained without the preliminary SCO_2_ treatment. After the first step the residual bark was extracted by microwave, ultrasound, or maceration using ethanol, ethanol/water 50%, or water as solvents. These extracts were compared with the one obtained with maceration, UAE or MAE, using the same solvents and same solvent mass: ratio but starting from untreated bark. After each extraction, solvent was removed by evaporation under vacuum when possible and the residual water by lyophilization. Nineteen different samples were obtained, and the process was compared considering percentual ponderal yield (weight dried extract/weight plant material) as well as phytochemical composition.

### 2.1. Ponderal Yields

SCO_2_ extraction was performed at 300 bar and at the temperature of 45 °C, yielding a clear yellow residue with a ponderal yield of 5%. Then, solvent based extractions were performed on the bark subjected to SCO_2_ and on the bark material, indicated in table as “RU” and “BU” respectively. After solvent removal by vacuum evaporation and lyophilization of water, the ponderal yields were calculated for each extract and are reported in Table 1.

We have observed some different behavior that can be explained due to three main variables, i.e., the SCO_2_, the chemical nature of the solvent used in the second extraction step, and the technique used in the second step. 

As example comparing the water yields of extraction, we can observe that in all the samples the pretreatment with SCO_2_ increase the amount of extracted material with the exception of microwave where the yields are very similar. This result can be explained due to the high efficiency of MAE when applied to water. Furthermore, we should keep in mind that the MAE is the only approach where the extraction is performed at hot temperature (70 °C), and thus the solubility of some of the constituents will be increased in water.

Considering the extraction step with the mixture water/ethanol for all the three extraction techniques (M, MAE, UAE) the higher yields are obtained after SCO_2_. This can be explained due to the improved solvent penetration in the plant material. M and MAE appear the more favorable. When extraction is performed only with ethanol, we observe similar behavior.

With this result we can state that the smart chain extraction procedure applied to the bark allows to obtain higher yield of extraction. This because as first result an SCO_2_ extract with 5% (*w*/*w*) of yield is obtained. From the same material, we can obtain yields of extracted materials comparable or higher than the one extracted with the procedures without the pre-treatment with SCO_2_. For exemplification, the best condition is the smart chain extraction with SCO_2_ (5%) and water/ethanol using microwave (4.79% in RMW_we) leading to nearly 10% yield compared to 3.30% obtained with water/ethanol using microwave (BMW_we). Furthermore, considering all the data, the yields of extraction are favorable when water/ethanol mixture is used as solvent. We can observe that in our case, after the extraction with SCO_2_, the most efficient solvent for the extraction of bark is the mixture ethanol/water 50% and comparing with the literature we found that the same solvent was previously reported as the more efficient for *Pinus pinaster* bark polyphenol extraction [7] where the same solvent was compared using Soxhlet apparatus.

### 2.2. ^1^H-NMR and LC-DAD-MS^n^ Analysis

^1^H-NMR was used to compare the composition of the SCO_2_ extract with the extracts obtained with solvents. ^1^H-NMR allowed to detect all the classes of constituents that present hydrogen atoms. This approach in this work was used as a screening technique to ascertain preliminary information concerning the composition of extracts. Due to the large number of different compounds, the spectra are poorly resolved but differences can be observed (Figure 2). SCO_2_ extract superimposed with UAE extract obtained with ethanol in Figure 2 show the different behavior. The SCO_2_ spectrum show signals ascribable to hydroxycinnamic esters, phenolics, and abietane diterpenes, as summarized in Table 2. Abietane diterpenes are a class of peculiar compounds that present some characteristic features in the ^1^H-NMR as previously reported in literature [8]. 

The signals presenting the same chemical shifts in the UAE extract are not detectable or poorly resolved, supporting the presence of different matrix. Results indicated that SCO_2_ extract presents different composition with large amount of abietane derivatives probably due to the lipophilic properties of the solvent. This suggests that SCO_2_ compared to solvent based extraction with ethanol and water/ethanol can selectively extract abietane and lipophilic hydroxycinnamic derivatives.

The ^1^H-NMR data, although very preliminary, allowed to observe that many secondary metabolites are present in the bark waste, and that the extraction methods used for the development of the “smart chain extraction” can efficiently extract these compounds from vegetal matrix. To obtain qualitative analysis of the different constituents we analyzed all the obtained extracts by LC-DAD-MS^n^. Due to our previous experience in bark constituents, we used our previously published method [9] allowing the identification of a series of phenolic, such as piceasides, luteolin, and taxifolin derivatives, as well as diterpene ascribable to abietic acid scaffold. The identification of compounds was achieved by the comparison with reference standard or isolated constituents as well as by the MS^n^ fragmentation data, identified compounds are reported in Table 3. SCO_2_ extract present high peaks ascribable to abietane diterpene and different behavior compared to other extracts in agreement with the ^1^H NMR data. All the solvent extracted samples, with M, UAE and MAE, presents comparable chromatographic profile at 280 nm, with broad signals between 12–1 min, and less intense signals between 24–36 min. The identified compounds can be grouped in several classes of phytoconstituents, organic acids (quinic, protocatechuic, ferulic, benzoic, ellagic and caffeic acid derivatives), flavonoid comprising flavanols, such as epi-catechin and procyanidin trimer, and taxifolin derivatives. Flavonol derivatives as isorhamnetin and quercetin, flavone as luteolin. Many stilbenoids have been detected both simple derivatives as piceatannol and trans-astringin, as well as oligomeric derivatives as piceasides. 7-hydroxy matairesinol was identified belonging to lignan class, and different diterpene mostly related to abietic acid. SCO_2_ extract was analyzed using the same methods and as attended the chromatogram presented the most intense signal in the region of abietane compound, and the most abundant compound was abietic acid.

### 2.3. Quantitative Analysis of Secondary Metabolites in P. abies Extracts

To compare the different extraction procedures, the amount of most significant secondary metabolites, namely abietane, piceasides, and flavonoids, was quantified by the LC-DAD method grouping the compound in the three main classes. Results are represented in histograms reported as mg/kg of starting material, namely the dried bark.

To evaluate the levels of secondary metabolites in the obtained extracts and to compare the efficacy of the extraction, the data were also expressed as mg/Kg of dried extract. We could observe that the SCO_2_ extract contains 8 g/Kg of abietanes, and only the extraction with ethanol from the untreated vegetal material allowed lower but similar extraction efficacy. This result suggests that SCO_2_ is ideal for the extraction of abietanes. The extracts obtained after SCO_2_ extraction presented very low amount of abietane showing the selectivity of SCO_2_ procedure. A significant increase of polyphenol extraction is observed for all the solvent-based extraction (UAE, MAE, M) when the process is performed after SCO_2_, indicating that the pre-treatment increases the polyphenol content in the final extract. Some differences are observed in the amount of the three classes of secondary metabolites due to the extraction technique. As a general trend, MAE and UAE allowed higher contents in secondary metabolites compared to M.

Looking at the obtained results and considering the environmental sustainability of the process, the smart chain extraction combining SCO_2_ and microwave water extraction allowed to obtain extract with significant amount of secondary metabolites without the use of organic solvents. Nevertheless, the ponderal yield (1.39% in RMW_w) is less than half compared to the microwave water/ethanol extraction (4.79% in RMW_we) that allow a similar content in secondary metabolites. In any case, smart chain extraction combining SCO_2_ and green solvent approaches looks attractive for extracting secondary metabolites from sawmill waste. Recently, sub and supercritical fluids extractions using ethanol water mixtures as modifiers were applied to *P. abies* bark. Two-step extraction was performed to separate lipophilic compounds from phenolics, showing efficient extraction [10].

### 2.4. Total Phenolic and Flavonoid Content in P. abies Extracts

Phenolic compounds are considered to represent one of the most important pharmaceutical and nutraceutical markers for the further uses of plant extracts [11]. With this in mind, we determined the total concentrations of phenolics and flavonoids using colorimetric methods. Clearly, the highest levels of total phenolics and flavonoids were determined in ethanolic and hydroalcoholic extracts. The highest value was found in the ethanolic extracts from residual bark in the maceration technique (125.82 mg GAE/gin RU_e). The extract also had the highest level of flavonoids with 7.85 g RE/g. In addition, the content of the total phenolics and flavonoids was increased by the smart extraction procedure with one exception (in ultrasound technique) (Figure 3). From this point, we concluded that the application of SCO_2_ in the first step could remove lipophilic substance from the cell membrane and this could facilitate the entry of polar solvents like ethanol or hydroethanol mixtures. SCO_2_ extract had the lowest levels of total phenolic (24.71 mg GAE/g) and flavonoid (0.31 mg RE/g) contents. In this sense, SCO_2_ extraction is not suitable to extract phenolics from bark. In the present study, the extracts obtained from green extraction techniques, namely UAE and MAE, contained more phenolics when compared to the traditional M technique. In a study conducted by Nisca et al. (2021), green extraction techniques (UAE and MAE) were suggested as effective to obtain more phenolics from *P. abies* [12]. Strižincová et al. (2019) studied on the optimization of phenolic compounds from *P. abies* via SCO_2_ extractions and the total phenolic level was ranged from 4.41 to 11.03 mg GAE/g [13], which was lower than that of our presented results. These different results could be explained by different locations of the plant samples and different parameters of the extraction equipment. As a concern with the spectrophotometric assay, different compounds can react with the reagent in different ways, and therefore the obtained result may not reflect the exact content in phenolics of a plant extract [14,15]. This can explain the differences that we obtained compared with the LC-DAD-MS data. Nevertheless, colorimetric assays due to their large applicability can be used as screening methods to assess the potential usefulness of extracts.

### 2.5. Antioxidant Properties of Analyzed Extracts

Different assays have been used to measure the antioxidant properties of the ninteen obtained extracts and the results are summarised in the Figure 4 (Appendix A). 

Antioxidant compounds are the key shields protecting cells from free radical onslaughts that lead to the progression of chronic and degenerative diseases [16]. In this context, we need to find new and safe sources of antioxidants especially related to potential application in nutraceuticals and cosmetics. Among the sources, plants are considered the most important treasure and contain various antioxidants, including phenolics, essential oils, and alkaloids. The current work investigated the antioxidant properties of *P. abies* extracts using different methods. Among the methods, DPPH and ABTS assays are radical quenching assays, and they are the most common assays in the phytochemical studies. From Figure 4, in both assays, the best free radical scavenging abilities were determined by the ethanol extracts of the residual bark obtained by maceration (DPPH: 604.36 mg TE/g and ABTS: 1225.75 mg TE/g in RM_e) and ultrasound techniques (DPPH: 600.91 mg TE/g and ABTS: 1053.89 mg TE/g in RU_e). For all extraction techniques, the tested ethanol and hydroalcoholic extracts showed stronger scavenging abilities compared to water extracts. The weakest extract was SCO_2_ in both assays. 

Reducing power assays, namely CUPRAC and FRAP, are closely related to electron-donation ability of the antioxidant molecules. If one extract has a great reducing power, the extract exhibits a good electron-donation ability and thus it will be considered as a good antioxidant. The results of reducing power assays are presented in Figure 5. In this set of measurements, the ethanol and water ethanol extract presented more favorable properties. It can be noticed that the extracts obtained with MW present generally better chelating properties compared to others.

With the exception of ultrasound extraction (water and ethanol extracts), the residual bark, namely from smart extraction, showed greater potential than the bark. The ethanol extract of residual bark (CUPRAC: 747.00 mg TE/g; FRAP: 496.95 mg TE/g in RM_e) in maceration technique was determined to be the most effective. The water extracts of bark and residual bark exhibited the lowest abilities in all extraction techniques. Once we evaluated the results of free radical quenching and reducing power assays together, the obtained results are consistent with the total phenolic content of the extracts. With this in mind, we concluded that phenolic compounds were key players in free radical scavenging and reducing power assays. These findings are consistent with those reported in the literature, where a positive correlation between total phenol and radical scavengers and reducing abilities has been reported [17,18]. The phosphomolybdenum assay also included the conversion of Mo (VI) to Mo (V) by antioxidants at acidic condition. In addition, since not only phenolics, but also non-phenolic antioxidants could play a role in the assay, it is known as a total antioxidant assay. Similar to CUPRAC and FRAP assays, the ethanol and hydroalcholic extracts exhibited greater potentials when compared with water extracts in the phosphomolybdenum assay. Unlike other assays, SCO_2_ wasn’t the weakest. This could be explained by the presence of non-phenolic reducing agents such as terpenoids. The chelation of transition metals, particularly iron, is an important antioxidant mechanism to control hydroxyl radical production. As can be seen from Figure 5, the best chelating ability was observed in the water extract of the bark in the microwave technique (9.91 mg EDTAE/g in BMW_w). In general, the results of metal chelating abilities were in contrast to other antioxidant assays. The different results could be explained by the presence of non-phenolic chelating agents such as peptides and polysaccharides. The close agreement of the results with the literature suggested that there was a weak correlation between metal chelation and total phenolic levels [19,20]. Several studies have been conducted in the literature to evaluate the antioxidant potential of *P. abies* from different countries. For example, Nisca et al. (2021) examined the antioxidant properties of *P. abies* extracts obtained from ultrasound and microwave extraction and the radical scavenging abilities of ultrasound extracts were higher than those of microwave extracts [12]. In addition, the authors reported a significant correlation between the total phenolic content of the tested *P. abies* extracts and the free radical scavenging abilities. Neiva et al. (2018) tested the antioxidant properties of the ethanol and water extracts of *P. abies* by DPPH and FRAP assays and the ethanol extracts showed stronger abilities in the assays compared to water extracts [21]. Their results are consistent with our results, where ethanol extracts had greater potential than water extracts. In another study by Angelis et al. (2016), the methanol fractions of *P. abies* bark showed good radical scavenging abilities (62.2%) at a low concentration (25 µg/mL) [22]. In addition to the studies, the observed antioxidant properties of *P. abies* extracts could be explained by the presence of some compounds in their chemical profiles, including piceaside, luteolin, and ellagic acid. Taken together, *P. abies* barks could be considered as a potential raw material with natural antioxidant properties in the development of health-promoting products.

### 2.6. Enzyme Inhibitory Properties

Over the past decade, the term enzyme inhibition has been one of the most popular in the scientific platform and refers to the treatment of some global health problems, including obesity, Alzheimer’s disease and type II diabetes. With this in mind, some compounds as enzyme inhibitors have been chemically produced and obtained from pharmacy shelves. However, most of these compounds have unfavorable side effects and need to be replaced with natural ones [23]. Plant secondary metabolites have great potential as natural enzyme inhibitors in the literature and studies in this area continue [24]. In light of this information, we tested the enzyme inhibitory properties of *P. abies* extracts against cholinesterases (AChE and BChE), tyrosinase, amylase, and glucosidase. The results are shown in Table 4. In both AChE and BChE inhibition assays, the best inhibitory abilities were determined in the ethanol and ethanol/water extracts. Most water extracts were less active than ethanol and ethanol/water. In addition, in general, the smart extraction technique was increased the observed the inhibitory properties for AChE and BChE. Tyrosinase is a key catalyst in the synthesis of melanin and its inhibition is a key way to control hyperpigmentation problems. In general, the best tyrosinase inhibitory effect was observed in the tested ethanol extracts. The highest tyrosinase inhibition value was determined in the ethanol extract of the residual bark by ultrasound technique. In the present study, similar findings were found for amylase inhibition and water extracts were the weakest of all extraction techniques. Interestingly, the best activity for amylase inhibition was recorded by SCO_2_ and this could be explained by the non-polar compounds like terpenoids, as SCO_2_ is more effective to extract non-polar compounds than polar ones. In glucosidase inhibition assay, most of the tested extracts were not active and the results contrast with other enzyme inhibitory assays. In the literature, few papers regarding enzyme inhibitory properties of *P. abies* have been found. In a study performed by Angelis et al. (2016), the tyrosinase inhibitory effects of *P. abies* extracts and fraction were reported [22]. In their study, the fraction showed a good inhibitory effect on tyrosinase as 51.7 and 58.7% at 100 µg/mL. In addition, the authors have been reported a good tyrosinase inhibitory activity of some isolated compounds form *P. abies* including taxifolin and astringin, the compounds had a great high tyrosinase inhibitory activity [22]. In this sense, in our presented study, the same compounds were detected and the observed tyrosinase inhibitory activity could be related to the presence of the compounds. In another study by Senol et al. (2015), the cholinesterase inhibitory effects of shoot and needle of *Picea pungens* and the extracts exhibited moderate inhibitory properties (AChE: 2.60–45.09% and BChE: 15.03–46.50% at 100 µg/mL) [25]. At this point, due to insufficient information on the enzyme inhibitory properties, our presented study might open new horizons for the utilization of *P. abies* extracts as sources of natural enzyme inhibitors that can be useful active materials for nutraceuticals purposes.

## 3. Discussion

Supercritical fluid extraction, compared to conventional methods, is a sustainable and cleaner technology that uses green, nontoxic, and nonflammable solvents that can be recycled for repeating the process. Furthermore, the SCO_2_ can be favorable to obtain extract with different composition compared to conventional approaches. As a further matter of advantage, the final extract has no residual solvents, being the ideal extraction approach to produce food grade ingredients, nutraceutical and natural derived products. Supercritical fluid technology was applied for the extraction of potential tropical biomass wastes for various types of applications, such as biopesticides, bio-repellents, phenolics, and lipids for biofuel, showing its role in circular bioeconomy and sustainable development approaches [26]. In this work, “smart chain extraction” was proposed starting from SCO_2_ to extract the lipophilic fraction of bark and proceeding with second step by solvent-based extraction to obtain extracts rich in polyphenols. Considering the obtained yields, the “smart chain extraction” approach resulted to be valuable both from ponderal yield point of view as well as considering the polyphenol concentration in the second step extraction. Considering ponderal yields (Table 1), “smart chain extraction” appeared favorable with all the solvent extraction approaches, and MAE ensured higher yields. Referring to amount of extracted secondary metabolites, the “smart chain extraction” appear favorable when the second step of extraction is performed with water/ethanol mixture. 

Recently other research work considering multistage extraction strategies were applied for the valorization of bark wastes. Multistage fractionation of pine bark was performed using subcritical and supercritical SCO_2_ at increasing pressures and temperatures [27]. Results revealed that after removing most of the lipophilic compounds in initials steps, the last supercritical extraction with ethanol as co-solvent, facilitates the recovery of more polar compounds, such as phenolics and glycerol, released through the depolymerization reactions of lignin and suberin [27]. Bento et al. (2022) presented a green strategy to sequentially extract *Pinus radiata* bark, exploring SCO_2_ extraction and a biocompatible ionic liquid catalyst. The obtained SCO_2_ extracts predominantly contained lipophilic constituents as resin acids [28]. Then, suberin was recovered preserving the high esterified polymeric backbone by using an ionic liquid-based extraction process [28]. Considering our work and comparing with the literature, the efficiency of SCO_2_ technique sequentially used in the valorization of natural compound in bark matrix is notable due to the high selectivity to specific classes of natural compounds. 

For the second extraction phase, we compared two innovative extraction approaches, UAE and MAE with classical maceration, showing that, in general, the first two approaches bring to improved yields and polyphenol concentration in the final extracts. Polyphenols extraction from pine bark were previously investigated using decoction [29], UAE [30,31,32] and MAE [33]. Using UAE, optimum extractions conditions were established using 70% ethanol as solvent, proving that the ethanol concentration was the most important variable, followed by time and temperature [31]. Recently, pine (*P. nigra* and *P. sylvestris*) barks were extracted with UAE and MAE to obtain extracts with antioxidant and antimicrobial activity. Results indicate that polyphenols were efficiently extracted from *P. nigra* bark with UAE while monoterpenes were better extracted in *P. sylvestris* bark with MAE [34], suggesting that extraction efficacy of the different methods can be specie specific. Multistage extraction with hot water extraction, slow pyrolysis, and anaerobic digestion was recently proposed for the extraction of polyphenols and biogas [35] from *P. abies* bark to find alternative and sustainable strategy for the valorization of this waste.

Previous work showed that pressurized fluid extraction resulted as an efficient technique to extract phenolic from *P. abies* bark. Authors reported that using water and ethanol as solvents for PFE at 160 and 180 °C yielded extracts with high antioxidant capacity. Stilbene glucosides, such as isorhapontin, piceid, and astringin, were identified as main constituents [36]. Other authors evaluated the opportunity to extract phenolics using SCO_2_ and co-solvent and optimal conditions were established using design of experiments as 73 °C, 44.5 MPa, and 58% EtOH/water cosolvent [16]. 

## 4. Materials and Methods

### 4.1. Plant Material

*P. abies* bark was kindly provided from Holz Pichler S.p.A. 8 kg of *P. Abies* bark was crushed using a RETSCH rotary blade mill GM-200 at a speed of 10,000 revolutions per minute for 1 min. 4 kg of bark powder was subjected to SCO2 extraction while 4 kg were directly used for the water-based extraction. Aliquots of 10g of bark powder were extracted with 125 mL of ethanol, ethanol-water 50%, and water by MAE, UAE and M. 

### 4.2. Chemicals and General Materials

Ethanol used for extraction and methanol, quercetin, abietic acid, polydatin, used for the analysis, were purchase from Sigma Aldrich (St. Louis, MO, USA).

### 4.3. SCO_2_ Extraction

Supercritical extraction of *P. abies* bark was performed with a TH22-10 x2 super-critical CO_2_ extraction apparatus (Toption Instrument Co., Ltd., Xi’an, China), as depicted in Figure 6. Briefly, the plant was equipped with two extraction vessels of 10 L and two separators of 5 L. The carbon dioxide (Siad SpA, Trieste, Italy; 99.99% purity, food grade) was carried with a high-pressure liquid pump (Toption Instrument Co., Ltd.).

First, 2.9 kg of milled *P. abies* bark (≤40 mesh) was weighed into the stainless-steel ex-traction basket, which was loaded onto the jacketed extraction vessel. The flow rate of supercritical solvent was set at 1 L/min in all experiments. The extraction pressure was set to 150 bar, while the extraction temperature was set at 40 °C. The first separator was operated at 70 bar and 45 °C and the second one at 45 bar and 40 °C. The extraction was carried on until the amount of extract collected over 1 h decreased to under 0.1% of the raw material. During the supercritical carbon dioxide extraction, water (bound moisture from plant material) was co-extracted, then decanted, and the crude extract was collected and stored. The crude extracts were weighed, and the yield was calculated as g extract/100 g dry material (d.m.). The extraction pressure and the flow were maintained constant using a backpressure regulator. The extraction led to SCO2 extract with ponderal yield of 5%.

### 4.4. Solvent Based Extraction

#### 4.4.1. Maceration (M)

Exactly weighted bark powder was transferred in three glass bikers and soaked with water, ethanol/water 50%, and ethanol respectively. Sample were mixed under magnetic stirrer for 30 min, at 25 °C. Extracts were filtered through cotton at first, and then through filter paper. Water extract was dried by lyophilization. Water/ethanol extract was partially dried to remove ethanol with a rotary evaporator. Then, a lyophilizator was used to remove residual water. Ethanol extract was dried using the rotary evaporator.

#### 4.4.2. Microwave-Assisted Extraction (MAE)

MAE was performed with an Ethos X Advance Microwave Extraction System (Milestone, Bergamo, Italy). The bark powder was transferred in the vessel of the instrument and soaked with water, ethanol/water 50% and ethanol, respectively for each extraction. Extraction parameters were power 200 W, 50 °C, for 18 min. Extracts were filtered through cotton at first, and then through filter paper. Extracts were dried as reported in Section 2.4.

#### 4.4.3. Ultrasound-Assisted Extraction (UAE)

UAE was performed in an ultrasound bath LABSONIC FALC- LBS1 (Treviglio, Italy). Exactly weighted bark powder was transferred in three glass flask and soaked with water, ethanol/water 50% and ethanol respectively. The samples were sonicated for 30 min at 200 W, filtered through cotton at first, and then through filter paper. Extracts were dried as reported in Section 2.4.

### 4.5. Calculation of Extraction Yield

Yields of extraction were calculated on the basis of dried material after SCO_2_ or solvent elimination obtained on weight of initial bark, and yields were expressed as % (*w*/*w*).

### 4.6. NMR Analysis

^1^H-NMR spectra were obtained on a Bruker Avance 400 spectrometer (Munich, Germany), using standard pulse sequences. SCO_2_ extract were prepared by dissolving the extract in deuterated chloroform (10 mg in 2 mL of solvent), while extracts obtained by UAE, MAE, and M were prepared in deuterated methanol (10 mg in 2 mL of solvent). Solutions were sonicated for 5 min and centrifuged for 5 min 10,000 rpm. Supernatant solutions were transferred in NMR tubes.

### 4.7. Liquid Chromatography–Diode Array Detector–Mass Spectrometry (LC-DAD-MS^n^)

Quali-quantitative analysis of secondary metabolites was conducted using the method previously published by our group [9]. LC was performed using a Agilent 1260 chromatograph equipped with 1260 diode array (DAD) and Agilent/Varian MS-500 ion trap (Santa Clara, CA, USA) as detectors. An Eclipse XDB C-8 2.1 × 150 mm 3.5 μm (Agilent, Santa Clara, CA, USA) column was used as stationary phase and acetonitrile (A) and 0.1% formic acid in water (B) were used as mobile phases. The elution gradient was set as follows: linear gradient from 90% B to 40% B, 0–45 min; linear gradient from 40% B to 0% B, 45–51 min; isocratic gradient 0% B, 51–55 min; linear gradient from 0% B to 90% B, 55–56 min, and isocratic gradient until 60 min. The flow rate was 0.3 mL/min and the injection volume was 10 μL. At the end of the column a T connector split the flow rate to DAD and MS detector. MS spectra were recorded in negative ion mode in 50–2000 Da range, using an ESI ion source. The turbo data depending scanning (TDDS) function allowed to obtain the fragmentation of the main ionic species. Identification of compounds was based on the fragmentation spectra, as well as the comparison of the fragmentation pattern with the literature and injection of reference compounds, when available. The DAD chromatograms were monitored at λ = 350, 330, 280, and 254 nm and were elaborated for the compound’s quantification. All flavonoids, phenols and piceasides were quantified using the external standard method. For the quantitative analysis, quercetin, abietic acid, and polydatin were used as reference standards for flavonoid and phenolics, abietan-type diterpenoid, piceasides and derivatives respectively. Calibration curves of the standards were prepared by diluting stock standard solutions in methanol to yield final concentrations in the range of 14.9–149 µg/mL for abietic acid, 14.5–145 µg/mL for polydatin, and 10.5–105 µg/mL for quercetin. Linear regressions were as follows: abietic acid y = 53,516x (R2 = 0.999); polydatin y = 81,290x (R2 = 0.999); quercetin y = 61,669x (R2 = 0.999). As sample preparation, 25 mg of the extracts obtained with MAE, UAE and M were dissolve in 30 of methanol, sonicated for 15 min and centrifuged for 5 min 10,000 rpm. 25 mg of SCO_2_ extract was solubilized in 20 mL of DMSO, sonicated for 15 min and centrifuged for 5 min 10,000 rpm. Solutions were used for analysis.

### 4.8. Total Phenolic and Flavonoid Content

The total phenolic and flavonoid contents were determined using the Folin-Ciocalteu and AlCl_3_ tests, respectively [37]. Results were presented as gallic acid equivalents (mg GAEs/g dry extract) and rutin equivalents (mg REs/g dry extract) for the assays.

### 4.9. Antioxidant Assays

Antioxidant assays were performed using methods that have been previously reported [38]. The antioxidant potential was calculated as follows: mg Trolox equivalents (TE)/g extract in the 2,2-diphenyl-1-picrylhydrazyl (DPPH) and 2,2′-azino-bis(3-ethylbenzothiazoline-6-sulfonic acid) (ABTS) radical scavenging tests; cupric reducing antioxidant capacity (CUPRAC) and ferric reducing antioxidant power (FRAP); in phosphomolybdenum assay (PBD) and mg ethylenediaminetetraacetic acid equivalents (EDTAE)/g extract in metal chelating assay (MCA).

### 4.10. Enzyme Inhibitory Assays

The enzyme inhibition experiments were performed based on previously described procedures [38]. Amylase and glucosidase inhibition was expressed as mmol acarbose equivalents (ACAE)/g extract, while acetylcholinesterase (AChE) and butyrylcholinesterase (BChE) inhibition was expressed as mg galanthamine equivalents (GALAE)/g extract. Tyrosinase inhibition was expressed as mg kojic acid equivalents (KAE)/g extract.

## 5. Conclusions

In this work, an innovative two-step extraction obtained using in sequence a first step with SCO_2_ and a second step using green solvents, namely ethanol, water, and water ethanol mixture was applied to *P. abies* bark, a sawmill residue, to extract valuable phytoconstituents. The results indicate that the initial step with SCO_2_ helps the second solvent-based extraction in increasing the total yield. Furthermore, the SCO_2_ extract due to its peculiar composition can be a valuable source of diterpenoids. In vitro bioassays revealed potential usefulness for the different extracts in cosmetic or nutraceutical applications thanks to significant antioxidant and enzyme inhibitory activities.

## Figures and Tables

**Figure 1 molecules-27-06719-f001:**
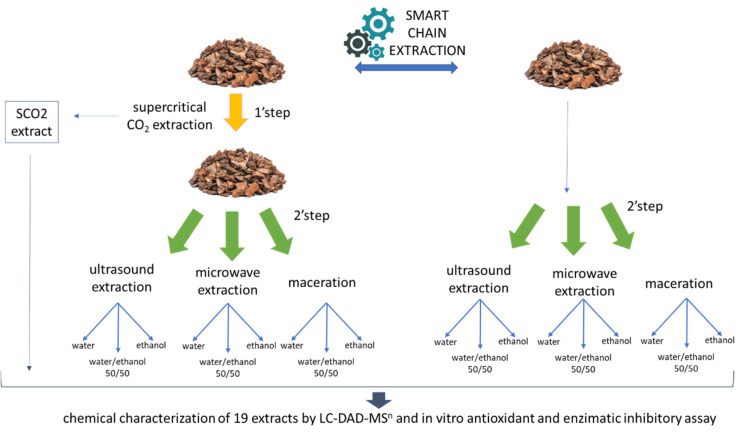
Scheme summarizing the different extraction processes studied in the paper.

**Figure 2 molecules-27-06719-f002:**
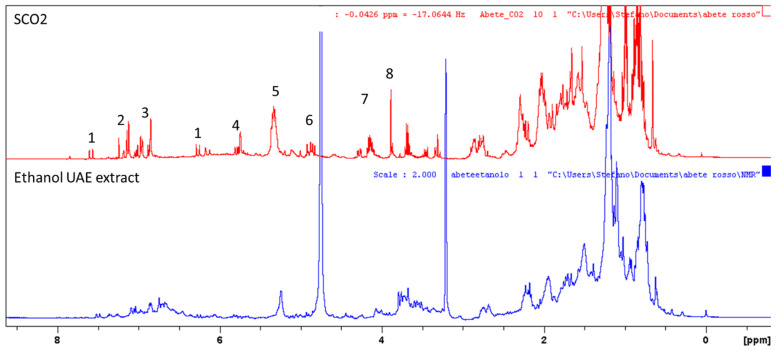
Main signals observed in the ^1^H-NMR spectrum of SCO_2_ and extract obtained from barks after UAE in ethanol (RU_e).

**Figure 3 molecules-27-06719-f003:**
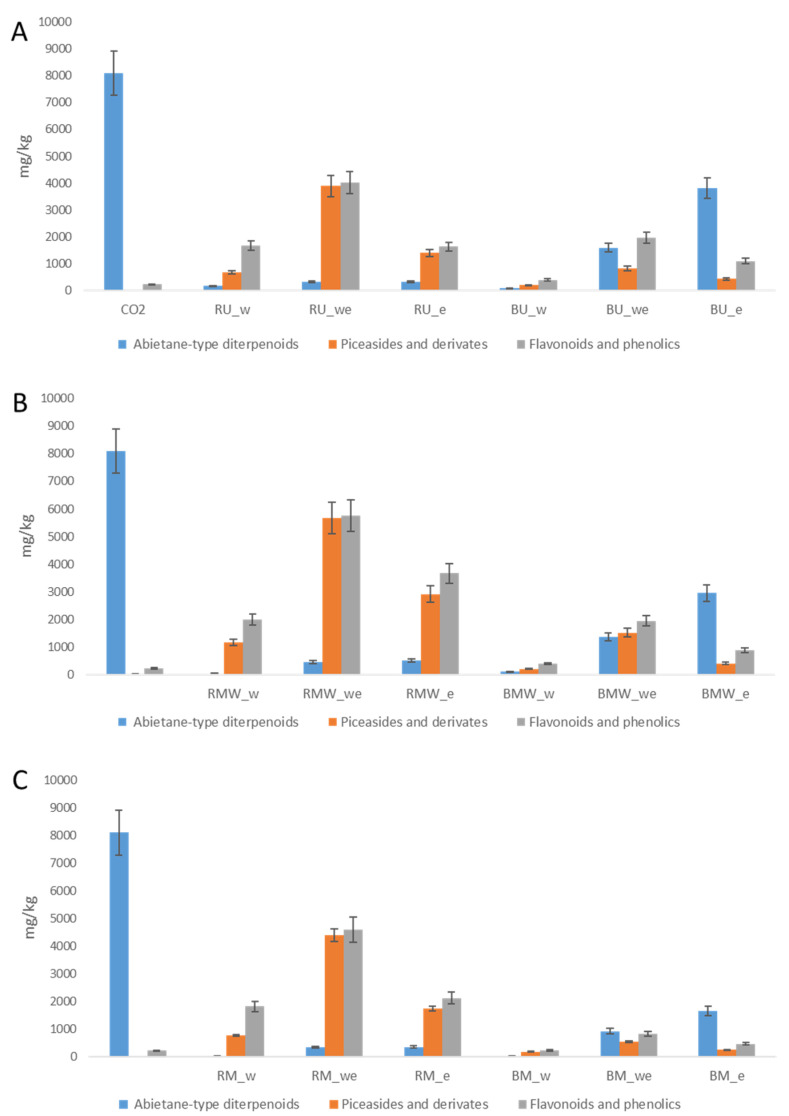
Amount of secondary compounds expressed as mg/Kg of bark obtained by UAE (**A**), MAE (**B**) and M (**C**). Values are reported in Appendix A.

**Figure 4 molecules-27-06719-f004:**
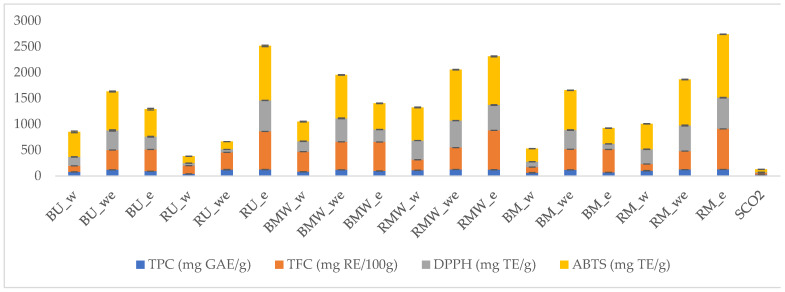
Total phenolic, flavonoid content, and free radical scavenging abilities of the tested extracts. Values are reported in Appendix A. Data are expressed as Gallic acid equivalents (GE) Rutin equivalents (RE) and Trolox equivalents (TE).

**Figure 5 molecules-27-06719-f005:**
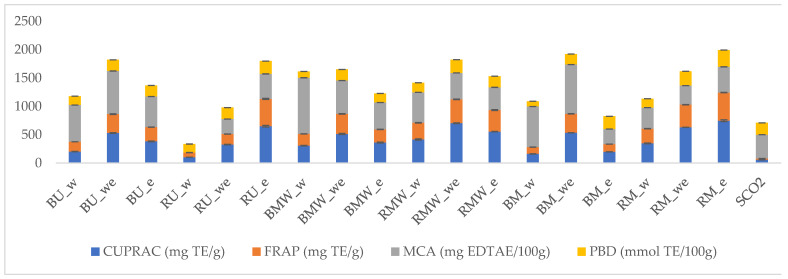
Reducing power (CUPRAC and FRAP), metal chelating (MCA) and total antioxidant capacity (by phosphomolybdenum assay (PBD)) of the tested extracts. Data are expressed as Trolox equivalent (TE) and EDTA equivalents (EDTA E).

**Figure 6 molecules-27-06719-f006:**
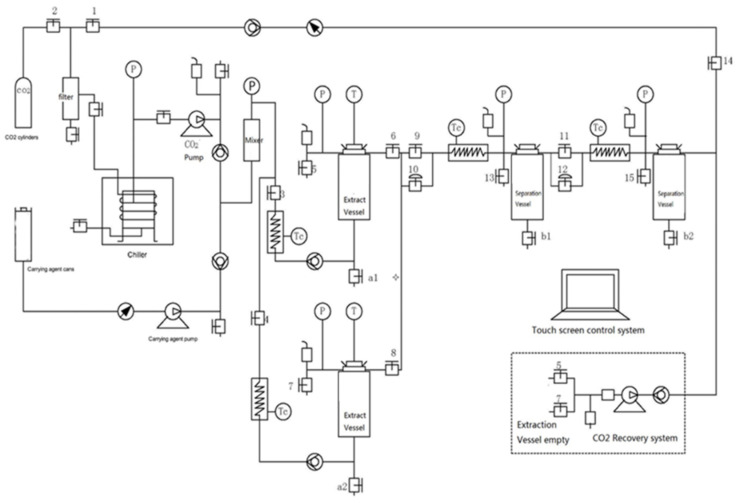
Supercritical extraction equipment. P = pressure controller, T = temperature controller, Tc = heater exchanger.

**Table 1 molecules-27-06719-t001:** Extraction yields and codification names of the nineteen extracts from *P. abies.*.

Extraction Technique	Starting Bark Material	Solvent	Sample	% Yield
UAE	bark	water	BU_w	1.07%
		ethanol/water 50%	BU_we	2.33%
		ethanol	BU_e	2.87%
	bark after SCO_2_	water	RU_w	2.78%
		ethanol/water 50%	RU_we	3.98%
		ethanol	RU_e	1.9%
MAE	bark	water	BMW_w	2.04%
		ethanol/water 50%	BMW_we	3.30%
		ethanol	BMW_e	2.59%
	bark after SCO_2_	water	RMW_w	1.99%
		ethanol/water 50%	RMW_we	4.79%
		ethanol	RMW_e	4.39%
M	bark	water	BM_w	1.48%
		ethanol/water 50%	BM_we	2.37%
		ethanol	BM_e	1.74%
	bark after SCO_2_	water	RM_w	2.63%
		ethanol/water 50%	RM_we	4.57%
		ethanol	RM_e	2.19%
SCO_2_	bark	SCO_2_	SCO_2_	5.00%

**Table 2 molecules-27-06719-t002:** Tentatively assignment of NMR signals ascribable to hydroxycinnamic esters, phenolics, abietanes.

Signal Number in Spectrum	δ H	Tentative Identification
1	7.60–6.45	Trans olefine signals of hydroxycinnamic acid derivatives
2	6.80–7.20	Aromatic signals ascribable to hydroxycinnamic derivatves, phenolics, flavonoids, stilbenoids
3	6.00–6.8	Aromatic signals ascribable to abietic acid derivatives or similar diterpene
4	5.60–5.80	Olefine of abietic acid derivatives ascribable to positions 7-8-13-14
5	5.20–5.40	Olefine signals of unsaturated fatty acids
6	5.20–4.80	Exocyclic sp2 olefine signals in diterpene derivatives
7	4.10–4.30	Oxigenated CH
8	3.80	Methoxy signal

**Table 3 molecules-27-06719-t003:** Identified constituents of *P. abies* bark extracts obtained by SCO_2_, UAE, MAE, M identified by HPLC-DAD-MS^n^.

Compound	[M − H]^−^ *m/z*	ESI-MS^n^ *m/z*
Hydroxy-piceaside derivative	665	485-443-305-243
Benzoic acid derivative	313	151-282
Caffeoyl-hexoside	341	203-179-131
Quinic acid *	191	127-111
Caffeic acid derivative	377	341-179
Procyanidin trimer B	865	695-577-407
Protocatechuic acid-hexoside	315	153-109
Ferulic acid *	193	173-145
(epi)-Catechin *	289	245-203
Hydroxy-piceaside derivative	665	485-443
Isorhamnetin *	315	299
Taxifolin-7-O-glucoside *	465	447-303-285
Luteolin-7-O-rhamnoside *	431	285-241
Hydroxy-piceaside derivative	665	485-443-305
Trans-astringin *	405	243
Hydroxy-piceaside derivative	665	503-445-297
Ellagic acid hexoside	463	301
Piceaside A/B	809	647-485-375
Hydroxy-piceaside derivative	665	503-445-297
Piceaside A/B	809	647-485-375-229
Isorhapontigenin	257	241-213
Piceatannol	243	225-201
Hydroxy-piceaside derivative	665	503-445-243
Piceaside A/B	809	647-485-375
Piceaside A/B	809	647-485-375-318
Piceaside G/H	809	646-405
Piceaside C/D	823	661-499
Piceaside C/D	823	661-499
Piceaside C/D	823	661-499
Piceaside G/H	809	646-405-243
Piceaside C/D	823	661-499-257
Taxifolin *	303	285-241-213
Isorhamnetin-pentoside	447	315-300
Piceaside E/F	823	661-499-241
7-hydroxy-matairesinol *	373	355-311-296
Piceaside G/H	809	646-405
Piceaside E/F	823	661-499-241
Piceaside G/H	809	646-405-243
Piceatannol derivative	647	485-243
Methoxy-piceatannol hexoside	661	499-241
Piceaside E/F	823	661-499-241
Methoxy-piceatannol	499	467-389-241
Quercetin *	301	179-151
Methyl abietate	315	301-257
Dehydroabietic acid	299	255
Abietic acid	301	257
12β hydroxy abieta 7-13 18 oic acid	333	289
7-Oxodehydroabietic acid	313	269
Abienol	289	191-163
13-Epi-manool	289	215

* identified by standard comparison

**Table 4 molecules-27-06719-t004:** Enzyme inhibitory properties of the tested extracts.

Sample	AChE (mg GALAE/g)	BChE (mg GALAE/g)	Tyrosinase (mg KAE/g)	Amylase (mmol ACAE/g)	Glucosidase (mmol ACAE/g)
BU_w	0.78 ± 0.05	1.70 ± 0.01	1.53 ± 0.09	0.05 ± 0.01	2.53 ± 0.01
BU_we	3.57 ± 0.03	3.37 ± 0.16	57.80 ± 0.67	0.40 ± 0.01	Na
BU_e	4.01 ± 0.07	4.30 ± 0.21	66.15 ± 1.10	0.37 ± 0.02	2.49 ± 0.01
RU_w	Na	4.60 ± 0.06	41.47 ± 0.58	0.19 ± 0.01	2.54 ± 0.01
RU_we	3.70 ± 0.04	3.71 ± 0.15	57.08 ± 0.45	0.41 ± 0.01	Na
RU_e	3.98 ± 0.03	4.63 ± 0.08	67.67 ± 0.37	0.33 ± 0.01	Na
BMW_w	0.52 ± 0.02	1.32 ± 0.07	2.80 ± 0.86	0.09 ± 0.01	Na
BMW_we	3.52 ± 0.06	3.04 ± 0.10	60.92 ± 0.75	0.39 ± 0.01	Na
BMW_e	3.75 ± 0.07	4.34 ± 0.23	63.11 ± 1.01	0.35 ± 0.01	2.35 ± 0.02
RMW_w	0.96 ± 0.03	2.25 ± 0.30	12.28 ± 0.76	0.05 ± 0.01	Na
RMW_we	3.69 ± 0.02	2.82 ± 0.09	60.36 ± 0.46	0.41 ± 0.03	Na
RMW_e	4.04 ± 0.03	4.77 ± 0.04	65.22 ± 0.66	0.32 ± 0.01	Na
BM_w	0.59 ± 0.11	0.76 ± 0.05	Na	0.05 ± 0.01	2.46 ± 0.01
BM_we	3.45 ± 0.07	2.64 ± 0.12	53.13 ± 0.91	0.33 ± 0.01	Na
BM_e	3.81 ± 0.08	4.54 ± 0.12	48.34 ± 1.00	0.35 ± 0.01	2.41 ± 0.01
RM_w	1.14 ± 0.06	2.52 ± 0.14	13.76 ± 0.73	0.05 ± 0.01	Na
RM_we	3.81 ± 0.01	3.68 ± 0.05	57.77 ± 0.20	0.32 ± 0.02	Na
RM_e	4.07 ± 0.04	4.76 ± 0.16	66.71 ± 0.35	0.29 ± 0.02	Na
SCO_2_	3.49 ± 0.10	4.50 ± 0.25	36.19 ± 0.76	0.45 ± 0.01	Na

Values are reported as mean ± SD of three parallel measurements. GALAE: Galanthamine equivalents: KAE: Kojic acid equivalents; ACAE: Acarbose equivalent. Na: not active.

## Data Availability

Not applicable.

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
