# Peer review of "“Smart Extraction Chain” with Green Solvents: Extraction of Bioactive Compounds from Picea abies Bark Waste for Pharmaceutical, Nutraceutical and Cosmetic Uses"

_molecules, 2022, doi:10.3390/molecules27196719_

Round 1

Reviewer 1 Report

The page I, the Introduction part, line 10, the word ‘whit’ is not clear

Table-1, percentage yield: write 1.07%, not 1,07%, do it everywhere in text

Write SCO2 everywhere in the text, instead of SCO2

Section 3.2: Write 1H-NMR

Table-4: I think the authors reported IC50 values of extract, so mention it. Give IC50 values of the standard drug also.

Write AlCl3 instead of AlCl3

Authors should provide LC-MS/MS spectra in supporting information.

Author Response

Review 1

The page I, the Introduction part, line 10, the word ‘whit’ is not clear

Thank for comment we revised

Table-1, percentage yield: write 1.07%, not 1,07%, do it everywhere in text

Thank for comment we revised

Write SCO2 everywhere in the text, instead of SCO2

Thank for comment we revised

Section 3.2: Write 1H-NMR

Thank for comment we revised

Table-4: I think the authors reported IC50 values of extract, so mention it. Give IC50 values of the standard drug also. Thank for comment we reported for all the test the value expressed as Gallic acid equivalents (GAE), Trolox Equivalents (TE) and EDTA equivalents (EA)

Write AlCl3 instead of AlCl3

Thank for comment we revised

Author Response

 Molecules Review Report

General comments

The manuscript should be thoroughly checked for typographical and grammatical errors as it is conspicuous all through the manuscript.

We thank for suggestion the manuscript was revised and checked. Blue colour highlight changes,

Results

The explanation of the data in Table 1 is quite poor and difficult to follow. Although it seems the water/ethanol extraction gave more yield the results need to be rewritten.

We thank for suggestion the description of table 1 was changed

The result on NMR and LC-DAD-MSn analysis is poorly explained. While it is evident that there are differences in the peaks, but the explanation is poor.

We thank for suggestion the description of table 1 was changed

In addition, the presentation of the graphs seems to busy, and I will suggest the authors look for better ways to represent them.

In a nutshell, the result section needs to be written so as to convey proper understanding of the manuscript.

We thank for suggestion we modified.

From the findings, CO2 alone did not give better yield but as a preliminary step before the use of other solvents, the yield was improved. Could this improved yield be attributed to the C02 or the solvents alone as there was no comparison showing the yields without C02.

We thank for the comment, all the data have been now better exposed, and we add a scheme that may clarify the aim and all the comparison of the different extractions.

Furthermore, could the antioxidant and enzymatic activities be linked to the intermediary C02 step, or they are based on the actions of the extracts alone irrespective of the intermediary CO2 step.

We thank for suggestion the manuscript was changed and we hope that the new version can be more clear and informative. The step with CO2 is not leaving any trace of the solvent thus there is no influence of the CO2 treatment on the activity of the obtained extracts, the activity is related to the extract composition.

Verdict.

I do not recommend the acceptance of this manuscript in this format. For it to be accepted it must be revised